# Towards a Domain-Agnostic Computable Policy Tool

Mitchell Falkow[0000−0001−7956−1182], Henrique Santos (✉)[0000−0002−2110−6416], and Deborah L. McGuinness[0000−0001−7037−4567]

Tetherless World Constellation, Rensselaer Polytechnic Institute, Troy NY, USA
{falkom,oliveh}@rpi.edu, dlm@cs.rpi.edu

**Abstract.** Policies are often crucial for decision-making in a wide range of domains. Typically they are written in natural language, which leaves room for different individual interpretations. In contrast, computable policies offer standardization for the structures that encode information, which can help decrease ambiguity and variability of interpretations. Sadly, the majority of computable policy frameworks are domain-specific or require tailored customization, limiting potential applications of this technology. For this reason, we propose ADAPT, a domain-agnostic policy tool that leverages domain knowledge, expressed in knowledge graphs, and employs W3C standards in semantics and provenance to enable the construction, visualization, and management of computable policies that include domain knowledge to reduce terminology inconsistencies, and augment the policy evaluation process.

## 1 Introduction

*Policies* (often referred to as *guidelines*) are sets of rules that describe the preferred responses to a given set of conditions, and typically are used to support practitioners with making decisions. Policies are often expressed using natural language. Natural language encodings are often ambiguous and sometimes incomplete, thus allowing for a potential range of interpretations. In contrast, computable policies (CPs) are typically written using frameworks that govern policy structures to ensure machine-readability, and allow for automatic evaluation using evaluation engine software. CPs are found in many contexts such as access control (e.g. XACML [1]) and healthcare (e.g. GEM [11], SAGE [14]).

Many frameworks are domain-specific, and using one outside its intended domain(s) often requires extensive configuration and/or the creation of customized extensions to the framework. Furthermore, the required changes can create unintended consequences, and sometimes errors.

One solution would be to make use of declarative domain knowledge expressed in an extensible and machine-understandable format stored within CPs. Besides preserving machine-readability and mitigating inconsistencies, this approach would enable the use of domain knowledge during policy evaluation. Very

---

Code and demo video: https://tetherless-world.github.io/adapt/

few frameworks are capable of using domain knowledge during evaluation, and of the frameworks that provide this capacity, even fewer provide user-friendly tools to help work within those frameworks.

For these reasons we introduce ADAPT, a domain-agnostic policy tool that leverages the domain knowledge stored in knowledge graphs (KGs). ADAPT represents CPs by extending KGs and by using recommended standards for ontologies (OWL) and provenance (PROV). Our solution enables the construction, visualization, and management of CPs that include machine-understandable domain knowledge. This approach can be used to reduce terminology inconsistencies and augment the policy evaluation process.

## 2   ADAPT Architecture

The ADAPT architecture (shown in Fig. 1) consists of a web-based user interface (UI), a RESTful backend API, and an RDF store that manages all policy and KG data. The role of the RDF store is fulfilled by the Tetherless World Knowledge Store[1].

Policies created in ADAPT use a generalization of the CP framework outlined in [9]. CPs that use this framework are capable of using existing OWL 2 reasoners to perform evaluation.

When creating policies, the UI makes HTTP requests to the backend API, which in turn uses SPARQL queries to extract attributes from the RDF store. The API processes the attributes and returns them as rule structures to the UI for use in policy construction. The API also wraps the standard HTTP method endpoints (e.g. GET, POST) for policies in the RDF store. These are used for tasks such as policy visualization and management.

ADAPT also aims to allow users to configure their domain by supplying their own domain KGs, and edit, browse, and visualize policies that they create.

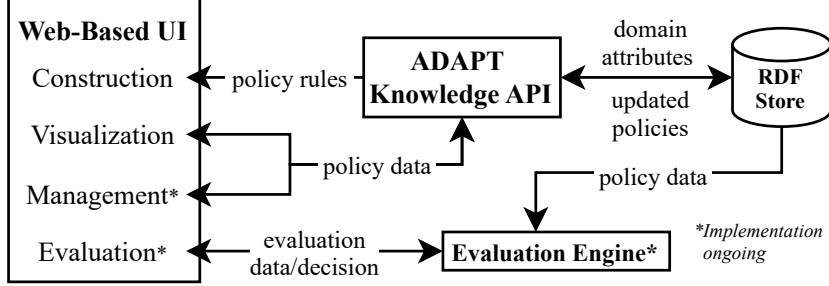

Fig. 1: ADAPT Architecture Diagram

---

[1] https://github.com/tetherless-world/twks

## 3 Demonstration: Healthcare Guidelines

We will illustrate the utility of ADAPT by creating a CP based on a guideline for diabetes patients from the American Diabetes Association [2]. Consider Recommendation 5.32, which reads:

> *Advise all patients not to use cigarettes and other tobacco products or e-cigarettes. (Evidence Rating = A)*

A policy author must interpret the policy, extracting the necessary information that composes the policy's rules. Policy *rules* are restrictions placed on PROV classes (`Activity`, `Entity`, `Agent`). The Semanticscience Integrated Ontology (SIO) [4] is used for attribute rules, which are restrictions defined with `sio:hasAttribute`. Terms from existing KGs can be reused by adding assertions.

Here, we assume a background KG where `Smoking`/`NotRecommended` and `Diabetes` are assumed to be subclasses of `prov:Activity` and `sio:Attribute`, respectively. We use `EvidenceRating` to denote policy precedence.

```
1 Class: Recommendation-5.32
2   EquivalentTo: SmokingCigarettes and (wasAssociatedWith some
3       (hasAttribute some Type1Diabetes))
4   SubClassOf: EvidenceRating-A, NotRecommended
```

Listing 1.1: Example Policy (Manchester Syntax)

To construct the policy (Listing 1.1), we first load the required KGs and definitions into the RDF store. ADAPT discovers and generates the valid rule structures. We then take these steps in the UI (see Fig. 2): (1) enter the *Source*, *ID*, *Label*, and *Description*; (2) specify *Smoking* for the action; (3) add a rule under *Rules* and select *Type 1 diabetes*; (4) select *Evidence Rating A* for *Precedence*; and lastly, (5) choose *Not Recommended* for the *Effect*. Once we click *Save*, ADAPT saves the policy to the RDF store and displays a graph view of the policy.

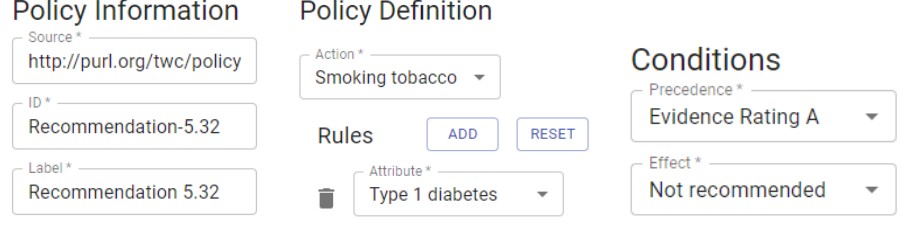

Fig. 2: Screenshots of ADAPT UI for Policy Construction

## 4 ADAPT and Other Computable Policy Tools

CP tools exist to aid in creating and editing policies. Many CP frameworks provide their own, as seen with GEM [5], SAGE [14]. Still these frameworks

are domain-specific, and while there is potential for reuse, significant changes would be needed from the user. In contrast, users configure ADAPT by providing domain knowledge from KGs. The benefit of this approach is that users can leverage domain knowledge from pre-existing domain KGs.

As for leveraging KGs, XACML does not readily provide this feature. Most frameworks in the medical domain make use of medical domain knowledge in the form of KGs [8], but users cannot necessarily supply their own domain KGs.

Torres et al. [13] propose a domain-independent tool for creating the medical equivalent of a CP. Similar to ADAPT, the tool uses KGs to support the authoring process. However, the resulting CP uses a domain-specific syntax Arden, so customization is needed for reuse, and reasoning via standard OWL reasoners would not be supported.

KAoS [15], Rei [6], and PROTUNE [3] are established frameworks that use Semantic Web technology. Approaches by Lopes et al. [7] and by Speiser et al. [12] are more recent Semantic Web approaches. Santos et al. [10] introduce a tool for building CPs in the dynamic spectrum access (DSA) domain, and uses the framework from [9]. ADAPT improves upon this tool by enabling usage in domains beyond DSA. Our choice for using the generalized framework in the latter is that it builds on previous works by matching the cross-domain policy expression semantics, enabling the implementation of a wide variety of attribute-based policies across domains.

## 5   Conclusion

We introduced ADAPT: a domain-agnostic tool for creating and visualizing computable policies that leverage domain knowledge from knowledge graphs. By employing Semantic Web standards, ADAPT permits users to create policies that maintain standards for terminology and augment policy evaluation by incorporating domain knowledge during the policy evaluation process.

The limitations of ADAPT in its current state include the dependency on users supplying ontologies that align with PROV and SIO, with limited flexibility. Santos et al. [10] combat this by allowing users to create terms on the fly. ADAPT should consider a similar approach going forward. Additional language support is necessary – currently ADAPT does not support the *union* of rules (e.g. `(Smoking ...) or Drinking`), which constrains the expressiveness of rules that can be created.

Finally, we are pursuing the incorporation of a policy evaluation engine, as indicated in Fig. 1, which will support the capacity to enforce policies.

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
