# OpenReview forum: "Towards a Domain-Agnostic Computable Policy Tool"
_eswc-conferences.org/ESWC/2021/Conference/Poster_and_Demo_Track — ESWC2021 P&D_

### Official Review · AnonReviewer1 · 2021-04-08
**Promising proposal but authors have to be clear about advantages, disadvantages and limitations**

**Rating:** 6
**Confidence:** 3

**Review:**

This paper presents a tool for defining computable policies based on Ontologies and Knowledge Graphs. Using these semantic technologies it can provide more flexibility across domains in comparison with state-of-the-art technologies. The presented tool is quite interesting and benefits and necessity arguments are well-grounded. However I have some concerns with the current version of the paper:

Firstly, you mention that a user can configure the domain incorporating domain-specific attributes from a domain-specific KG to the Knowledge Store. Doing that, we can dynamically change the aimed domain in the tool. However, depending on many factors (such as the broadness of the persisted knowledge) we can end up with non-complex enough policies. Therefore, users should enrich this content to fulfil their requirements, and thus, removing part of the system flexibility. Do you really think that this system can be flexible enough in comparison with other mentioned domain-specific approaches?

Another related concern is about the last sentence of the paper: "at supporting broader reuse in a wide range of domains around loading and reuse of knowledge graphs". So, is the system not capable to deal with Knowledge Graph heterogeneity nowadays? In addition, in the video you say around minute 6:30 that "it will take any set of ontologies, hopefully". Why hopefully? Is it not so flexible? Then, if it is not so flexible which is the final advantage for users?

In Fig. 2, I would suggest to include the full capture (if there is space in the final version) because it would really help to understand the policy definition procedure.

In Section 4, I do not understand why Protege is brought up into the discussion. I see Protege as a general tool which indeed can be used for this task but is not convenient at all.

So, in summary it is an interesting proposal but you really need to be clear about advantages, disadvantages and current limitations. Also, you have to sketch your future work so it is clear how current limitations could be solved.

Some typos per section:

Introduction:

allowfor automatic -> allow for automatic

Most framework do not readily provide this capacity, and those that do lack tools to help work within those frameworks -> (Rephrase, it is not clear what you are saying in the last part of the sentence)

Section 2:

Figure 1. -> Fig 1. (follow your own abbreviations)

a knowledge store to store [...] knowledge store is fulfilled by the Tetherless World Knowledge Store -> (too much store in two sentences, rephrase)

by supplying their own domain KGs, and edit browse, and visualize policies that they create -> (rephrase)

Section 4:

Santos et al. [6] introduces -> Santos et al. [6] introduce

Section 5:

broader reuse in a wide range of domains around loading and reuse of knowledge graphs -> (rephrase)

**Anonymity:**

Yes, I would like my review to remain anonymous.

---

### Official Review · ~Sabrina_Kirrane2 · 2021-04-14
**The Proposal tackles an important topic that is of interest to the community**

**Rating:** 8
**Confidence:** 5

**Review:**

This paper proposes a domain agnostic computable policy tool that leverages semantic technologies in order to model both policies and domain knowledge.

The paper is well written, the supporting video is easy to follow, and the proposal tackles an important topic that is of interest to the community.

Although the paper focuses on the architecture and the tool, I would be interested in knowing more about the policies:

- For instance, how does the proposal compare to general policy languages, such as Rei, Protune, and KAoS.
- What is the expressiveness of the language? Could it also be used to specify privacy preferences, licenses, contracts, etc?
- Does your work focus on policy specification and visualisation, or are you also interested in enforcement, compliance and governance? If yes you man be interested in the following:

Havur, G., Vander Sande, M. and Kirrane, S., 2020. Greater Control and Transparency in Personal Data Processing. In International Conference on Information Systems Security and Privacy (ICISSP).

Bonatti, P.A., Kirrane, S., Petrova, I.M. and Sauro, L., 2020. Machine Understandable Policies and GDPR Compliance Checking. KI-Künstliche Intelligenz,

De Vos, M., Kirrane, S., Padget, J. and Satoh, K., 2019, September. ODRL policy modelling and compliance checking. In International Joint Conference on Rules and Reasoning

**Anonymity:**

No, I would like my review to be deanonymized.

---

### Official Review · AnonReviewer2 · 2021-04-16
**A UI and an API to define and adapt policies**

**Rating:** 6
**Confidence:** 3

**Review:**

The submission presents ADAPT - a tool to define and adapt policies. The submission is accompanied by a video that demonstrates the UI for different data sets loaded into the back-end. The tool seems to be *useful* to author and maintain policies.

Policies are a long-standing topic in research around semantic web topics, so the work is highly *relevant* to the conference.
Regarding *topicality* and *originality*, I do not give full points, as the topic are around for some time and to have a UI and an API to edit policies is something that one would expect to exist. If it were embedded into some bigger approach where the UI showcases particularities of the approach, that would change my verdict to something more enthusiastic.

I wonder about the architecture. Why does it take the API if it seemingly only wraps SPARQL queries? What do the different colours, boxes, and arrows in Figure 1 mean?

Although this is of course not a research paper, I wonder why the authors only cite policy work from around 2000 and then their own work from 20 years later. Looking at the headlines, some things that may be relevant from in between:
* Lopes, Kirrane, Zimmermann, Polleres, Mileo: A Logic Programming approach for Access Control over RDF. ICLP 2012
* Speiser and Studer: A Self-Policing Policy Language. ISWC 2010



**Anonymity:**

Yes, I would like my review to remain anonymous.

---

### Decision · Program_Chairs · 2021-04-19

Accept